# Incidence and Risk Factors Affecting the Recurrence of Primary Retinal Detachment in a Tertiary Hospital in Spain

**DOI:** 10.3390/jcm11154551

**Published:** 2022-08-04

**Authors:** Cristina Irigoyen, Ainhoa Goikoetxea-Zubeldia, Jorge Sanchez-Molina, Asier Amenabar Alonso, Miguel Ruiz-Miguel, Maria Teresa Iglesias-Gaspar

**Affiliations:** 1Donostia University Hospital, 20014 San Sebastian, Spain; 2Medicine Department, University of the Basque Country (EHU/UPV), 48940 San Sebastian, Spain; 3Division of Neurosciences, Biodonostia Health Research Institute, 20014 San Sebastian, Spain; 4Clinical Epidemiology, Biodonostia Health Research Institute, 20014 San Sebastian, Spain; 5CIBERESP ISCIII, Carlos III Health Institute, 28029 Madrid, Spain

**Keywords:** retinal detachment, recurrence, risk factors, incidence

## Abstract

(1) Objective: To determine the incidence, visual outcomes and risk factors associated with the recurrence of primary retinal detachment (RD) in a tertiary hospital. (2) Methods: A retrospective observational study was conducted, and data were collected on all eyes diagnosed with primary RD between January 2017 and December 2020. A detailed database was generated with data on anatomic and visual outcomes, and surgical technique information, for all the cases. (3) Results: 570 eyes with primary RD were included. Mean annual incidence of primary RD was 21.8 cases per 100,000 inhabitants. Mean follow-up time was 465 (±410.5) days. Mean time to redetachment was 114.4 (±215.8) days, with the median being 35 days. Statistically significant variables related to a higher risk of recurrence were: male sex (*p* = 0.04), type of tamponade (*p* = 0.01), surgeon (*p* = 0.035), inferonasal (*p* = 0.002) and inferotemporal (*p* = 0.032) involvement, complex RD (*p* < 0.001) and ocular comorbidity (*p* < 0.001). More satisfactory final visual acuity (VA) in patients not suffering redetachment was associated with shorter duration of central vision loss. (4) Conclusions: Sex, type of tamponade, inferior detachment, RD complexity, surgeon and ocular comorbidity were identified as prognostic factors for recurrence. Worse final postoperative VA was found in patients referring central vision loss for more than 4 days before surgery.

## 1. Introduction

Retinal detachment (RD) refers to the separation of the neurosensory retina from the retinal pigment epithelium below, which causes fluid accumulation between the two layers. Depending on the causal mechanism or the pathogenesis, four different types of RDs are defined: rhegmatogenous (the most common), tractional, exudative and combined rhegmatogenous/tractional [1,2].

Even with the correct diagnosis and the best treatment, the prognosis varies considerably between patients, as does the probability of successful primary surgery. 

Causes of surgery failure mentioned in early publications include the extent of the detachment, the number of breaks, lens status (mainly aphakia), myopia, inflammation, retinal and choroidal atrophy, both vitreous and retinal haemorrhage, vitreous loss and a failure to identify all retinal breaks [3,4]. Some years later, in an article published by Rachal and Burton [5], more emphasis was placed on an incorrect surgical technique and other factors that prevented the adequate closure of all the retinal tears. From the analysis of 1088 cases, they identified the following factors related to failure (ranked by frequency): massive preretinal retraction (now known as proliferative vitreoretinopathy (PVR)), preretinal membrane, undetected retinal tears, inadequate scleral buckle, new retinal tears, inadequate chorioretinal reaction, iatrogenic retinal tears, loss of buckle height and macular hole. Inadequate closure of existent retinal breaks accounted for 77% of primary surgical failures.

At present, PVR is considered one of the most important factors in primary surgical failure [1,6,7,8], being associated with a lower success rate, ranging between 45% and 85% [8,9,10]. Williamson et al. found 22% lower success rates in the presence of PVR. It is also worth noting that as the grade of PVR increases, so does the risk of surgical failure. Furthermore, in patients with PVR, inferonasal (IN) or posterior breaks and 4 quadrants RD are considered risk factors for redetachment, while the break position being temporal or superior is protective. On the other hand, in patients without PVR, it is difficult to determine the variables that may influence success; in this context, it is extremely important to decide the best surgical option based on the retinal characteristics observed before surgery [11].

The main objective of the study was to identify the incidence of retinal redetachment after primary surgery and potential associated risk factors. The secondary objective was to analyse variables that affect the probability of achieving final visual acuity (VA) of ≤0.30 logMAR.

## 2. Materials and Methods

This is a retrospective observational study in which medical data were collected on all eyes diagnosed with primary RD between January 2017 and December 2020 at Donostia University Hospital. Patients with complex RD, high myopia and other characteristics often used as exclusion criteria were included in the study; however, secondary RDs were excluded. Therefore, we have analysed the entire population treated for primary RD at a reference tertiary hospital. A total of 570 cases was included.

Inclusion criteria included patients with primary RD during the study period. We collected data for patients with anatomical failure after primary RD surgery (secondary RD following successful primary surgery, or when RD was persistent due to surgical failure), as due to the postoperative follow-up protocol of patients in our hospital we were not able to differentiate between patients with persistent RD related to primary anatomical failure and patients suffering RD recurrence. Exclusion criteria included patients with previous RD surgery.

Information was obtained retrospectively from patient electronic medical records and entered into Microsoft Excel. Variables studied included: sex, age, laterality, lens status, myopic refractive error, preoperative VA, duration of central vision loss, macular status, number of breaks, extent of RD, inferior break if present, ocular comorbidities, presence of PVR, complex RD, surgical technique, surgeon, type of tamponade, recurrence and date of recurrence if present, VA (at the last visit) and length of follow- up until discharge. RD was considered complex in the following situations: chronic RD (≥3 months), total RD, traumatic RD, previous glaucoma surgery, PVR, proliferative diabetic retinopathy, current or previous uveitis, vitreous haemorrhage, high myopia (≥−6D), giant break (≥90), coexistence of macular hole, accompanying choroidal detachment, retinal dialysis, complex cataract, endophthalmitis and/or presence of any other ocular infection or tumours.

All the surgeries were performed by experienced vitreoretinal surgeons at the same hospital. The surgical technique was chosen at the surgeon’s discretion.

All data were analysed using IBM SPSS Statistics 28.0. For continuous variables, results are presented as means, standard deviation (SD), median and range, while for categorical variables numbers (n) and percentages have been used. First, a descriptive analysis of the data was carried out to explore the characteristics of the patients. The effects of each variable on redetachment and final VA were assessed using univariate chi-square analysis tests for categorical variables and Student’s t tests for continuous variables. A *p* value of <0.05 was considered statistically significant. After that, logistic regression was executed comparing no redetachment and redetachment groups as well as those considered to have satisfactory final VA (≤0.30 logMAR) and unsatisfactory final VA (>0.30 logMAR). For each risk factor, odds ratios and 95% confidence intervals were defined. 

The hospital’s Ethics Committee approved the study. All research adhered to the Declaration of Helsinki.

## 3. Results

Over the 4 years of the study, the mean population studied was 652,862 and primary RD surgery was performed in a total of 570 cases, yielding an average annual incidence of 21.8 cases per 100,000 population (Table 1). By type, the average annual incidence rates were 21 cases per 100,000 for rhegmatogenous RD, 0.7 cases per/100,000 for tractional RD and 0.1 cases per/100,000 for exudative RD.

Table 2 shows the demographic characteristics and surgical techniques of eyes included in the study.

Eyes were followed up for a mean time of 465 (±410.5) days. Recurrence occurred in 28.9% (n = 165) of patients. For each group, the specific redetachment rate was 28.6% for rhegmatogenous RD, 44.4% for tractional RD and 0% for exudative RD. The average time between surgery and recurrence was 114.4 (±215.8) days, with a median of 35 days (Table 3).

In univariate analysis, statistically significant results in relation to primary RD recurrence were found for the following variables: sex (*p* = 0.04), detachment in the inferior-temporal (IT) (*p* = 0.032) or the inferior-nasal (IN) quadrant (*p* = 0.002), ocular comorbidities (*p* < 0.001), complex RD (*p* < 0.001), type of tamponade (*p* = 0.01) and surgeon (*p* = 0.035). Specifically, the rate of recurrence of RD was 21.6% in women and 33.1% in men. Further, the redetachment rate was 32.9% if the IT quadrant was involved, versus 24.7% if this quadrant was undamaged, and 36.2% if the IN quadrant was involved, versus 24.1% if it was unaffected. Lastly, redetachment was seen in 34.7% of cases with complex RD, and 36.8% of those with ocular comorbidities, while in the absence of these two risk factors, recurrence was seen in 20.5% of cases.

The most common causes of complex RD (>80 cases) were high myopia, PVR and vitreous haemorrhage, while among complex RD, the most frequent causes of recurrence (>45% recurrence rate) were endophthalmitis, previous glaucoma surgery, choroidal detachment, uveitis, giant break and complex cataract (Table 4). Ocular comorbidities were found in 269 patients, 79.5% of whom were considered complex RD. In the remaining 20.5% of patients with ocular comorbidities, the most prevalent ones were age-related macular degeneration, cataract and ocular hypertension. The most common ocular comorbidities were high myopia, glaucoma, epiretinal membrane, amblyopia and ocular hypertension. Regarding tamponade, the recurrence rate varied with the technique used: 25.5% with SF6, 27.3% with air, 33.7% with C3F8 and 44.4% with silicone oil (SO). No statistically significant relation was found when comparing different concentrations of each gas tamponade. Table 5 shows the results of the univariate analysis.

Forty-five percent of patients were pseudophakic. The mean time from cataract surgery to retinal detachment was 3.4 years (median of 2 years).

In the univariate analysis exploring variables potentially associated with redetachment, no significant relation was found for the following factors: age, laterality, lens status, pre-operative VA, myopia, duration of central vision loss, macular status, number of breaks, inferior breaks, superior-nasal (SN) and superior-temporal (ST) involvement, PVR, surgical technique, use of laser, use of cryotherapy, scleral buckling or sub-retinal fluid drainage through retinotomy.

PPV was the first-line treatment in 556 cases, while scleral buckle was used in 39 cases. The recurrence rates associated with PPV and scleral buckle were 28.98% and 25.6%, respectively, the difference not being statistically significant (Table 2). Further, regarding surgical techniques, the addition of scleral buckling to 23G pars plana vitrectomy (PPV) or 25G PPV was not found to be significant.

After multivariate analysis and logistic regression with the aforementioned variables, sex and ocular comorbidities remained statistically significant, but not IT or IN involvement, surgeon, complex RD or type of tamponade (Table 6). Specifically, being male increased the risk of recurrence 2-fold (*p* = 0.002) and having ocular comorbidities by 1.9-fold (*p* = 0.015).

In univariate analysis for the impact of variables on final postoperative VA in patients not suffering redetachment, we found statistically significant differences for: older age (*p* < 0.001), myopic refractive error (*p* < 0.001), type of tamponade method (SF6, C3F8 and air achieving better VA than SO, *p* < 0.001), presence of ocular comorbidities (*p* < 0.001), duration time from VA loss to surgery (*p* = 0.004), detached macula (*p* < 0.001), number of breaks (*p* < 0.001), IT (*p* = 0.004) and IN involvement (*p* < 0.001), existence of inferior retinal breaks (*p* = 0.003), development of PVR (*p* < 0.001), complex RD (*p* = 0.001), use of cryotherapy (*p*= 0.01) and sub-retinal fluid drainage through retinotomy (*p* < 0.001). We did not find any association with lens status (Table 7).

Time from progressive central vision loss to surgery was one week or more in 29.8% and more than two weeks in 14.5% of cases (Table 5)

Figure 1 shows the percentage of eyes in the group of non-recurrence macula-off RD with final VA ≤ 0.30 logMAR correlated with the duration of central vision loss prior to surgery.

## 4. Discussion

The main objective of our study was to evaluate the annual incidence of primary RD in our setting and the factors influencing surgical failure. We found an incidence of 21.8 cases per 100,000 inhabitants per year. This is similar to the rate found in a study in Denmark [12]. Other studies in Caucasian populations (including primary RD regardless of type) have found higher [13] and lower [14] incidences of primary RD than in our region.

Recurrences occurred in 28.9% of cases; considering that the mean time to recurrence was 114.42 days, with a median of 35 days, it is important to emphasize the need for close monitoring of the patient during the first 5 weeks.

Pars plana vitrectomy (PPV) was the first-line treatment in 556 cases, while scleral buckle was used in 39 cases. The recurrence rates associated with PPV and scleral buckle were 29% and 25.6%, respectively, the difference not being statistically significant. These results highlight the recent changes in trends in RD surgery, favouring with a growing tendency to PPV over scleral buckle as the preferred surgical technique [15]. In a Cochrane library systematic review comparing PPV and scleral buckle for repairing simple rhegmatogenous detachment, the authors found a rate of redetachment after PPV of 21%, lower than our results [2]. This can be explained if we take into account that they only evaluated outcomes in cases of simple RDs, while no exclusion criteria were applied in our study. Indeed, nearly two-thirds (65%) of our cases of recurrence were in eyes classified as having complex RD. Excluding cases of complex RD and those with ocular comorbidities, the recurrence rate in our sample drops to 20.5%, similar to what has already been reported [2]. Our rates of recurrence after scleral buckle surgery were similar to the current published literature [2]. Other studies have also found lower recurrence rates with PPV, although it is important to note that they excluded traumatic, tractional and exudative RDs [2,16,17].

PVR is the main cause of redetachment described in the current literature [1,6,7,8]. This study did not find PVR as a statistically significant risk factor for RD recurrence. Only 14.4% of all cases presented with PVR, and the recurrence rate in this group was 32.9%. These results might be underestimated due to the retrospective design of the present study.

In our study, a considerable number of patients were classified as having complex RD, including those with macular hole (n = 18), complex cataract (n = 26), previous filtering glaucoma surgery (n = 13), chronic RD (n = 9), traumatic RD (n = 43), giant retinal tear (n = 21), total RD (n = 42) and vitreous haemorrhage (n = 82). Some of these factors have been shown to increase the risk of surgical failure, and this may explain our higher recurrence rates [18,19,20,21,22,23].

We observed higher rates of recurrence when using SO as tamponade. In total, 44.4% of retinas showed redetachment after the intraocular injection of SO intraocular injection. Notably, 91.7% of procedures in which SO was used were classified as complex RD; this may explain the high rate of recurrences. Stanley et al. evaluated the efficacy of SO in complex RD, finding rates of complete attachment that varied between 70% and 78% [24], while Scott et al. reported redetachment rates at 1 year after the injection of 1000 or 5000 centistokes SO of between 18% and 21% [25]. Even though these studies included secondary surgery after a first recurrence, which likely explains the differences in attachment rates seen in our study, the factors underlying our results still remain to be clarified.

In our study, the recurrence rate was significantly associated with inferior sector involvement, with redetachment being observed in over a third (34.3%) of these cases. This is a strikingly high result in comparison with the rest of the medical literature. In most cases, our patients underwent PPV, with a smaller percentage being treated with scleral buckling than in other series [15,26]. Some studies have suggested that scleral buckle surgery has some advantages in inferior RD [27,28]. Our low rates of buckling in primary RD procedures (7% of cases) could help explain our redetachment rates in eyes with inferior involvement.

Nearly half of the patients in this study were pseudophakic, with a mean time of 3.4 years from cataract surgery to retinal detachment (median of 2 years). Evidence of increased risk of RD after cataract surgery can be found in the literature, although time between cataract surgery and RD differs between studies. Risk of pseudophakic RD is considered to be higher in the first 6–24 months after cataract surgery, with the greatest risk during the first year. Some factors such as posterior capsule rupture or the surgery being made by a trainee surgeon have been shown to shorten the median time from surgery to pseudophakic RD. However, after the first 2 years, the risk of pseudophakic RD still remains higher for a decade [29]. Therefore, it would be advisable to inform the patients of a higher risk of RD after surgery, especially during the first years, and explain the visual symptoms associated with this pathology in order to achieve an early ocular assessment.

No retinal tears were found in 60 eyes (10.5% of cases) and among these, redetachment occurred in 22 cases (36.7%). This result might be overestimated due to the retrospective design of the study, because some clinical and surgical reports may lack the total numbers of retinal tears found at the surgery. It has been reported in the literature that the primary success rate decreases to 75% when no break is found [11]; this underlines the importance of devoting sufficient time to both diagnosis and the training of surgeons. In cases in which no break is found, the surgical success rate does not differ between PPV combined with scleral buckling and scleral buckling alone [30].

After multivariate analysis and logistic regression, the two variables associated with redetachment in a statistically significant manner were sex and ocular comorbidities. In our study, 63.5% of patients were male. The recurrence rate in male patients was 33.1% versus 21.6% in female patients. We identified male sex as a risk factor for recurrence after primary RD surgery (OR = 2, IC = 1.4–5.7). Gerstenberger et al. found higher incidence rates of rhegmatogenous RD in males in a population-based cohort study (OR = 4.16) [13]. Furthermore, the published literature has also identified higher rates of reoperation in males than females; Callaway et al. found that women were less likely than men to undergo reoperation after primary PR (OR 0.7, 95% CI 0.62, 0.79, *p* < 001) [31]. The reason underlying these results could be due to biological factors, such as abnormal adhesions in the vitreoretinal interface and longer axial lengths seen in males [31]. Additionally, behaviour differences between sexes could interfere with adherence to postoperative positioning [32]. Future studies will have to be carried out to analyse the influence of sex on retinal redetachment.

Regarding the surgeon, the recurrence rate in our study varied from 16% to 39.7%. We did not find statistically significant differences in surgical technique between surgeons; this might be due to our small sample. Surgeons with the lowest recurrence rate were surgeon number 3 (16% recurrence rate), surgeon number 1 (20%) (who performed only 25 procedures during the 4 years) and surgeon number 5 (24.3%). Surgeon number 3 had completed a vitreoretinal fellowship training, and surgeon number 5 had more professional experience on RD surgery during their lifetime. The three surgeons with less recurrences use non-contact viewing systems for vitrectomy, compared to the contact viewing systems used by the rest of the surgeons. Recently, the PRO study reported no differences in anatomic success between non-contact and contact viewing systems [33]. Mazinani et al. examined the relationship between experience and success rates and observed that primary anatomical success rate became stable after 500 vitreoretinal procedures. However, none of the surgeons in this study reached 500 procedures in the period studied and there are no data of the total number of procedures they have performed throughout their professional careers [34].

The long period associated with visual symptoms before diagnosis reported by some of our patients merits comment. Notably, 29.8% of the eyes had experienced progressive central vision loss for one week or more before surgery, while as many as 14.5% underwent surgery after 2 weeks with central visual loss. This highlights the need to create care pathways for referral, integrating primary care and ophthalmology specialists, to improve early detection of RD in our population.

Furthermore, it is essential to increase awareness of the symptoms of RD in the population. Longer reported duration of central vision loss in macula-off RD was associated with a poorer functional outcome, hence the importance of shortening this period in patients seeking medical attention. As can be seen in Figure 1, in non-recurrent macula-off RD, the final VA tends to be lower when central vision loss is experienced for more than 4 days before surgery. This affirms the findings of Yorston et al. and emphasises the need for surgery within 72–96 h [35]. Grabowska et al. also affirms the importance of re-thinking our priorities in RD surgery, giving more importance to macula-off RD repair within 3 days [36], contrary to previous articles, which suggested less urgency when repairing macula-off RD [37].

One of the strengths of our study was the lack of exclusion criteria used in other studies on the topic, such as complex RD. Given this, it reflects the real population that experiences primary RD. We believe that this increases the external validity of the study findings. As for the limitations of the study, one of the most serious was its retrospective nature, as this led to difficulties in collecting some data. For example, we did not include the axial length of the patients and its relationship with risk of redetachment. Another limitation of the study was that, due to the postoperative follow-up protocol of patients in our hospital, we were not able to differentiate between patients with persistent RD related to primary anatomical failure and patients suffering RD recurrence. Therefore, we decided to include all cases with anatomical failure after primary RD surgery, based on the guidelines developed by the Australian New Zealand Society of Retinal Surgeons [38].

## 5. Conclusions

In conclusion, incidence and recurrence rates were comparable in this article with the current literature. The main risk factors for retinal detachment were male sex, SO tamponade, inferior RD, surgeon, ocular comorbidities and complex RD. The two only independent variables associated with redetachment after logistic regression were male sex and ocular comorbidities. It is advisable to audit our results in order to inform our patients about the risk of recurrence in retinal detachment surgery. Worse final postoperative VA was found in patients referring central vision loss for more than 4 days before surgery. Not only urgency in surgery but the prompt referral of patients to vitreoretinal services should be our goal to try to reduce the days of central vision loss and improve the visual outcome of our patients.

## Figures and Tables

**Figure 1 jcm-11-04551-f001:**
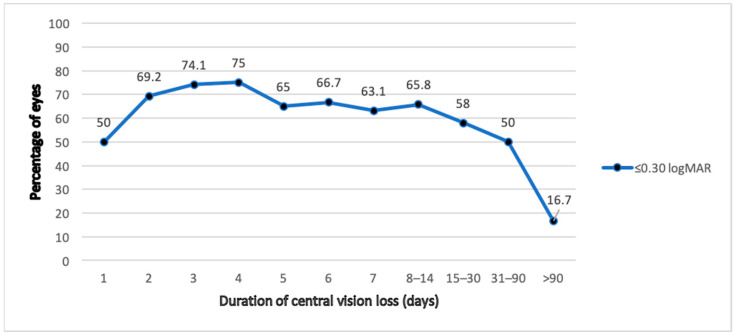
Relation between duration of central vision loss and percentage of eyes with final visual acuity of ≤0.30 logMAR, in non-recurrent macula-off RD.

**Table 1 jcm-11-04551-t001:** Annual incidence and 4-year mean incidence.

Year	Population Examined (n)	Primary RD Cases (n)	Annual Incidence Per 100,000 Inhabitants
2017	649,075	148	22.8
2018	651,201	162	24.8
2019	654,027	147	22.5
2020	657,145	113	17.2
Mean population studied = 652,862 Total primary RD cases = 570 Incidence = 21.8 cases per 100,000 per year			

**Table 2 jcm-11-04551-t002:** Characteristics of the study population expressed as numbers (n) and percentages (%).

Characteristics		n (%)
**Sex**	Female	208 (36.5)
	Male	362 (63.5)
**Age (years)**	<50	103 (18.1)
	50–69	306 (53.7)
	70–79	106 (18.6)
	≥80	55 (9.6)
**Laterality**	Right	310 (54.4)
	Left	260 (45.6)
**Lens status**	Phakic	298 (52.3)
	Pseudophakic	256 (44.9)
	Aphakic	8 (1.4)
	IOL phakic	8 (1.4)
**Pre-operative visual acuity (logMAR)**	≤0.30	178 (31.2)
	<1.00–>0.30	105 (18.4)
	≤1.30–≥1.00	80 (14.0)
	CF–NPL	196 (34.4)
**Post-operative visual acuity (logMAR)**	≤0.30	333 (58.4)
	≤1.00–>0.30	116 (20.4)
	>1.00	108 (18.9)
**Myopia**	No	188 (33.0)
	<3 Diopters	80 (14.0)
	3–6 Diopters	49 (8.6)
	>6 Diopters	100 (17.5)
**Ocular comorbidity**	No	299 (52.5)
	Yes	269 (47.2)
**Duration of central vision loss**	0	216 (37.9)
	1	4 (0.7)
	2	15 (2.6)
	3	34 (6.0)
	4	41 (7.2)
	5	35 (6.1)
	6	36 (6.3)
	7	24 (4.2)
	8–14	59 (10.4)
	15–30	44 (7.7)
	31–90	26 (4.6)
	>90	9 (1.6)
**Macular status**	On	254 (44.6)
	Off	308 (54.0)
**Number of breaks**	0	60 (10.5)
	1	289 (50.7)
	2	120 (21.1)
	3	48 (8.4)
	4	23 (4.0)
	5	6 (1.1)
	6	4 (0.7)
	7	1 (0.2)
	8	0 (0.0)
	9	1 (0.2)
**Quadrants involved**	ST	329 (57.7)
	IT	289 (50.7)
	SN	238 (41.8)
	IN	224 (39.3)
**Inferior breaks**	No	363 (63.7)
	Yes	195 (34.2)
	No data	12 (2.1)
**Proliferative vitreoretinopathy**	No	488 (85.6)
	Yes	82 (14.4)
**Complex retinal detachment**	No	262 (46.0)
	Yes	308 (54.0)
**Surgical technique**	23G PPV	128 (22.5)
	25G PPV	428 (75.1)
	Scleral buckle surgery	10 (1.8)
**Laser**	No	134 (23.5)
	Yes	435 (76.3)
**Cryotherapy**	No	349 (61.2)
	Yes	220 (38.6)
**Scleral buckle**	No	530 (93.0)
	Yes	39 (6.8)
**Type of tamponade**	SF6	373 (65.4)
	C3F8	83 (14.6)
	Silicone oil	72 (12.6)
	Air	22 (3.9)
**Sub-retinal fluid drainage**	Break	516 (90.5)
	Retinotomy	53 (9.3)
**Redetachment**	No	405 (71.1)
	Yes	165 (28.9)

**Table 3 jcm-11-04551-t003:** Characteristics of the expressed as mean, standard deviation, median, minimum and maximum.

Characteristics	Mean	Standard Deviation	Median	Minimum	Maximum
**Age (years)**	61.14	14.29	62	0	95
**Follow up (days)**	465	410.55	360	23	1800
**Years from cataract surgery to RD**	3.44	4.43	2	0	25
**Days from primary RD surgery to re-detachment**	114.42	215.78	35	1	1225

RD: retinal detachment.

**Table 4 jcm-11-04551-t004:** Relationship of each type of complex retinal detachment (RD) with recurrence. The table summarises all features of the methodology identified as complicating factors. The number (n) and percentage (%) of cases of no redetachment and redetachment within each characteristic are indicated, as are the total numbers of patients with each characteristic.

Cause of Complex RD	No Redetachment n(%)	Redetachment n(%)	Total
**High myopia (≥−6D)**	66 (66.0)	34 (34.0)	100
**Proliferative vitreoretinopathy**	55 (67.1)	27 (32.9)	82
**Vitreous haemorrhage**	59 (72.0)	23 (28.0)	82
**Traumatic RD**	30 (69.8)	13 (30.2)	43
**Total RD**	25 (59.5)	17 (40.5)	42
**Complex cataract**	14 (53.8)	12 (46.2)	26
**Choroidal detachment**	11 (47.8)	12 (52.2)	23
**Giant break (≥90),**	11 (52.4)	10 (47.6)	21
**Macular hole**	11 (61.1)	7 (38.9)	18
**Proliferative diabetic retinopathy**	9 (60.0)	6 (40.0)	15
**Previous glaucoma surgery**	6 (46.2)	7 (53.8)	13
**Retinal dialysis**	10 (83.3)	2 (16.7)	12
**Chronic RD (≥3 months)**	7 (77.8)	2 (22.2)	9
**Uveitis**	4 (50.0)	4 (50.0)	8
**Endophtalmitis**	1 (33.3)	2 (66.7)	3

**Table 5 jcm-11-04551-t005:** Univariate analysis of the variables used for retinal redetachment analysis. Univariate analysis of qualitative variables that might be associated with retinal redetachment was performed using chi-square tests. The number and percentage of cases of no redetachment and redetachment within each variable are indicated, as are the total numbers for each variable. Chi-square *p*-values are shown, considering *p* < 0.05 to be statistically significant.

Variable		No Redetachment n(%)	Redetachment n(%)	Total	*p* Value
**Sex**	Female	163 (78.4)	45 (21.6)	208	0.04
	Male	242 (66.9)	120 (33.1)	362	
**Age (years)**	<50	71 (68.9)	32 (31.1)	103	0.170
	50–69	227 (74.2)	79 (25.8)	306	
	70–79	74 (69.8)	32 (30.2)	106	
	≥80	33 (60.0)	22 (40.0)	55	
**Laterality**	Right	211 (68.1)	99 (31.9)	310	0.086
	Left	194 (74.6)	260 (25.4)	260	
**Lens status**	Phakic	214 (71.8)	84 (28.2)	298	0.920
	Pseudophakic	180 (70.3)	76 (29.7)	256	
	Aphakic	5 (62.5)	3 (37.5)	8	
	Phakic intraocular lens	6 (75.0)	2 (25.0)	8	
**Pre-operative VA (logMAR)**	≤0.30	136 (76.4)	42 (23.6)	178	0.232
	<1.00–>0.30	73 (69.5)	32 (30.5)	105	
	≤1.30–≥1.00	57 (71.3)	23 (28.7)	80	
	CF–NPL	131 (66.8)	65 (33.2)	196	
**Post-operative VA (logMAR)**	≤0.30	279 (83.8)	54 (16.2)	333	<0.001
	≤1.00–>0.30	79 (68.1)	37 (31.9)	116	
	>1.00	37 (34.3)	71 (65.7)	108	
**Myopia**	No	137 (45.5)	51 (44.0)	188	0.157
	≤3 Diopters	57 (71.3)	23 (28.7)	80	
	3–6 Diopters	41 (83.7)	8 (16.3)	49	
	≥6 Diopters	66 (66.0)	34 (34.0)	100	
**Ocular comorbidity**	No	233 (77.9)	66 (22.1)	299	<0.001
	Yes	170 (63.2)	99 (36.8)	269	
**Duration of central vision loss (days)**	0	157 (72.7)	59 (27.3)	216	0.300
	1	2 (50.0)	2 (50.0)	4	
	2	13 (86.7)	2 (13.3)	15	
	3	28 (82.4)	6 (17.6)	34	
	4	25 (61.0)	16 (39.0)	41	
	5	20 (57.1)	15 (42.9)	35	
	6	27 (75.0)	9 (25.0)	36	
	7	19 (79.2)	5 (20.8)	24	
	8–14	41 (69.5)	18 (30.5)	59	
	15–30	32 (72.7)	12 (27.3)	44	
	31–90	16 (61.5)	10 (38.5)	26	
	>90	7 (77.8)	2 (22.2)	9	
**Macular status**	On	189 (74.4)	65 (25.6)	254	0.124
	Off	211 (68.5)	97 (31.5)	308	
**Number of breaks**	0	38 (63.3)	22 (36.7)	60	0.485
	1	211 (73.0)	78 (27.0)	289	
	2	87 (72.5)	33 (27.5)	120	
	3	33 (68.8)	15 (31.2)	48	
	4	16 (69.6)	7 (30.4)	23	
	5	5 (83.3)	1 (16.7)	6	
	6	4 (100)	0 (0)	4	
	7	1 (100)	0 (0)	1	
	8	0	0	0	
	9	0 (0)	1 (100)	1	
**Quadrands involved**	ST no	164 (68.6)	75 (31.4)	239	0.261
	ST yes	240 (72.9)	89 (27.1)	329	
	IT no	210 (75.3)	69 (24.7)	279	0.032
	IT yes	194 (67.1)	95 (32.9)	289	
	SN no	235 (71.2)	95 (28.8)	330	0.958
	SN yes	169 (71.0)	69 (29.0)	238	
	IN no	261 (75.9)	83 (24.1)	344	0.002
	IN yes	143 (63.8)	81 (36.2)	224	
**Inferior breaks**	No	263 (72.5)	100 (27.5)	363	0.499
	Yes	136 (69.7)	59 (30.3)	195	
**Proliferative vitreoretinopathy**	No	350 (71.7)	138 (28.3)	488	0.390
	Yes	55 (67.1)	27 (32.9)	82	
**Complex RD**	No	204 (77.9)	58 (22.1)	262	<0.001
	Yes	201 (65.3)	107 (34.7)	308	
**Surgical technique**	23G PPV	91 (71.1)	37 (28.9)	128	0.825
	25G PPV	304 (71.0)	124 (29.0)	428	
	Classic surgery	8 (80.0)	2 (20.0)	10	
**Laser**	No	99 (73.9)	35 (26.1)	134	0.401
	Yes	305 (70.1)	130 (29.9)	435	
**Cryotherapy**	No	239 (68.5)	110 (31.5)	349	0.095
	Yes	165 (75.0)	55 (25.0)	220	
**Scleral buckle**	No	375 (70.8)	155 (29.2)	530	0.632
	Yes	29 (74.4)	10 (25.6)	39	
**Tamponade method**	SF6	278 (74.5)	95 (25.5)	373	0.01
	C3F8	55 (66.3)	28 (33.7)	83	
	Silicone oil	40 (55.6)	32 (44.4)	72	
	Air	16 (72.7)	6 (27.3)	22	
**Sub-retinal fluid drainage**	Break	370 (71.7)	146 (28.3)	516	0.248
	Retinotomy	34 (64.2)	19 (35.8)	53	
**Surgeon**	1	20 (80)	5 (20)	25	0.035
	2	51 (63)	30 (37)	81	
	3	68 (84)	13 (16)	81	
	4	55 (66.3)	28 (33.7)	83	
	5	81 (75.7)	26 (24.3)	107	
	6	90 (70.9)	37 (29.1)	127	
	7	38 (60.3)	25 (39.7)	63	

**Table 6 jcm-11-04551-t006:** Logistic regression. After multivariate analysis of the variables that might be associated with primary RD redetachment, results are shown for variables found to be significant. Results are represented as odds ratios, 95% confidence intervals and *p* values.

Variable		Odds Ratio	*p* Value	95% Confidence Interval
**Sex**	Female	Reference	-	-
	Male	2	0.002	1.4–5.7
**Ocular comorbidity**	No	Reference	-	-
	Yes	1.9	0.015	1.2–2.9

**Table 7 jcm-11-04551-t007:** Univariate analysis of the variables used for final post-operative VA analysis. Univariate analysis of qualitative variables that might be associated with final post-operative VA in patients not suffering redetachment was performed using chi-square tests. The number and percentage of cases of VA > 0.3 logMAR and VA ≤ 0.30 logMAR within each variable, as well as the total number for each variable, were indicated. Chi-square *p* values are shown, considering *p* < 0.05 to be statistically significant.

Variable		VA > 1.0 logMAR n(%)	VA 1.0–0.3 logMAR n(%)	VA ≤ 0.30 logMAR n(%)	Total	*p* Value
**Age (years)**	<50	5 (7.7)	18 (26.5)	45 (66.2)	68	<0.001
	50–69	13 (5.8)	32 (14.3)	178 (79.8)	223	
	70–79	9 (12.5)	17 (23.6)	46 (63.9)	72	
	≥80	10 (31.3)	12 (37.5)	10 (31.3)	32	
**Laterality**	Right	18 (8.6)	42 (20.1)	149 (71.3)	209	0.861
	Left	19 (10.2)	37 (19.9)	130 (69.9)	186	
**Lens**	Phakic	16 (7.6)	37 (17.6)	157 (74.8)	210	0.351
	Pseudophakic	20	38	116	174	
	Aphakic	1 (20.0)	2 (40.0)	2 (40.0)	5	
	IOL phakic	0 (0.0)	2 (33.3)	4 (66.7)	6	
**Myopia**	No	7 (5.3)	24 (18.0)	102 (76.7)	133	<0.001
	≤3 Diopters	3 (5.4)	11 (19.6)	42 (75.0)	56	
	3–6 Diopters	0 (0.0)	1 (2.4)	40 (97.6)	41	
	≥6 Diopters	13 (20.0)	15 (23.1)	37 (56.9)	65	
**Ocular comorbidities**	No	7 (3.0)	39 (17.0)	184 (80.0)	230	<0.001
	Yes	30 (18.4)	40 (24.5)	93 (57.1)	163	
**Days of central vision loss**	0	14 (9.1)	17 (11)	123 (79.9)	154	0.004
	1	1 (50.0)	0 (0.0)	1 (50.0)	2	
	2	0 (0.0)	4 (30.8)	9 (69.2)	13	
	3	2 (7.41)	5 (18.5)	20 (74.1)	27	
	4	0 (0.0)	6 (25.0)	18 (75.0)	24	
	5	2 (10.0)	5 (25.0)	13 (65.0)	20	
	6	4 (14.8)	5 (18.5)	18 (66.7)	27	
	7	1 (5.3)	6 (31.6)	12 (63.1)	19	
	8–14	2 (4.9)	12 (29.3)	27 (65.8)	41	
	15–30	3 (9.7)	10 (32.3)	18 (58.0)	31	
	31–90	2 (12.5)	6 (37.5)	8 (50.0)	16	
	>90	3 (50.0)	2 (33.3)	1 (16.7)	6	
**Macular status**	On	13 (7.0)	20 (10.7)	153 (82.3)	186	<0.001
	Off	24 (11.7)	56 (27.3)	125 (61.0)	205	
**Number of breaks**	0	16 (44.4)	5 (13.9)	15 (41.7)	36	<0.001
	1–4	18 (5.3)	68 (19.9)	255 (74.8)	341	
	5–9	0 (0.0)	3 (30.0)	7 (70.0)	10	
**Involvement**	ST no	18 (11.3)	28 (17.6)	113 (71.1)	159	0.427
	ST yes	19 (8.1)	50 (21.3)	116 (70.6)	235	
	IT no	10 (4.8)	43 (20.6)	156 (75.6)	209	0.004
	IT yes	27 (14.6)	35 (18.9)	123 (66.5)	185	
	SN no	19 (8.3)	42 (18.3)	169 (73.5)	230	0.374
	SN yes	18 (11.0)	36 (21.9)	110 (67.1)	164	
	IN no	11 (4.3)	52 (20.2)	195 (75.6)	258	<0.001
	IN yes	26 (19.1)	26 (19.1)	84 (61.8)	136	
**Inferior breaks**	No	32 (12.4)	53 (20.5)	173 (67.1)	258	0.003
	Yes	3 (2.3)	25 (18.9)	104 (78.8)	132	
**PVR**	No	24 (7.0)	59 (17.2)	260 (75.8)	343	<0.001
	Yes	13 (25.0)	20 (38.5)	19 (36.5)	52	
**Complex RD**	No	10 (5.0)	34 (16.8)	158 (78.2)	202	0.001
	Yes	27 (14.0)	45 (23.3)	121 (62.7)	193	
**Surgical technique**	23G PPV	11 (12.4)	19 (21.3)	59 (66.3)	89	0.543
	25G PPV	26 (8.8)	57 (19.2)	213 (72.0)	296	
	Classic surgery	0 (0.0)	3 (33.3)	6 (66.7)	9	
**Laser**	No	4 (4.2)	20 (20.8)	72 (75.0)	96	0.130
	Yes	33 (11.1)	59 (19.8)	206 (69.1)	298	
**Cryotherapy**	No	30 (12.9)	41 (17.7)	161 (69.4)	232	0.01
	Yes	7 (4.3)	38 (23.5)	117 (72.2)	162	
**Scleral buckle**	No	33 (8.9)	73 (19.7)	265 (71.4)	371	0.248
	Yes	4 (17.4)	6 (26.1)	13 (56.5)	23	
**Tamponade method**	SF6	12 (4.4)	53 (19.4)	208 (76.2)	273	<0.001
	C3F8	1 (1.8)	12 (22.2)	41 (76)	54	
	Silicone oil	24 (64.9)	9 (24.3)	4 (10.8)	37	
	Air	0 (0.0)	1 (6.2)	15 (93.8)	16	
**Sub-retinal fluid drainage**	Break	28 (7.8)	70 (19.4)	262 (72.8)	360	<0.001

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
