# Peer review of "Incidence and Risk Factors Affecting the Recurrence of Primary Retinal Detachment in a Tertiary Hospital in Spain"

_jcm, 2022, doi:10.3390/jcm11154551_

Round 1

Reviewer 1 Report

In this study, Irigoyen et al. evaluated incidence and risk factors regarding recurrence of primary RD in Spain. This study is well-written. Although the topic or the findings are not new, they are still important. 

My only comment is regarding ocular co-morbidities, which seem to be included as a collective dichotomous variable, i.e. either any presence or no presence. This is unfortunate, as further eloboration on this topic could potentially give more insight. Further, there are no data on the degree of myopia, any information of refraction in pseudophakic patients, or any information on axial length (e.g. post-operative). Information on the risk of re-detachment and relationship to axial length/myopia would be informative. Regardless, since this is a retrospective study and such information are rarely collected systematically, this is likely not any data which the authors could provide anyway. However, the lack of such data potentially influences the certainty regarding the results (e.g. how would they affect the currently significant variables when included in a multivariate analysis). And this limitation needs to be acknowledged in the manuscript.

Author Response

Dear reviewer,

Thank you for your comments. Regarding your review, we will answer point by point:

  1. Regarding ocular comorbidities, we have added the following sentence: "Ocular comorbidities were found in 269 patients, 79,5% of whom were considered complex RD. In the remaining 20,5% of patients with ocular comorbidities the most prevalent ones were age related macular degeneration, cataract and ocular hypertension."
  2. Regarding the myopia and axial length, unfortunately we did not measure the axial length of our patients due to the retrospective design of the study. The degree of myopia was classified in tables 2, 5 and 7. We did not find statistically significant results for the degree of myopia regarding retinal re-detachment.
  3. We have acknowleged as a limitation of our study what you suggested: "As for the limitations of the study, one of the most serious was its retrospective nature, as this has led to difficulties in collecting some data. For example, we did not include the axial length of the patients and its relationship with risk of redetachment. "

Kind regards.

Reviewer 2 Report

The authors reported the incidence, visual outcomes and risk factors associated with recurrence of primary retinal detachment (RD) in a tertiary hospital in Spain. They have identified tons of risk factors but after logistic regression, the only two independent variables associated with re-detachment were male and ocular comorbidities, which are pretty obvious to us already. I don’t see any additional insight this study could bring to the readers.

The other major concern is that the authors considered the persistent RD due to surgical failure as a recurrence, which sounds unreasonable.

Author Response

Dear reviewer, 

Thank you for your comments. We would like to reply to them as follows:

  1. Regarding the additional insight that our paper could bring to the readers, we consider that our study shows that macula-off RD repair should be considered more urgent than previous beliefs, as cited in the conclusions:

"Worse final postoperative VA was found in patients referring central vision loss for more than 4 days before surgery. Not only the urgency in surgery but the prompt referral of patients to the vitreoretinal service should be our goal to try to reduce the days of central vision loss and improve the visual outcome of our patients."

In the literature, the repair of macula-off RD was considered less urgent than the period we found in our study. As we have now included in the resubmission of our study, recent published literature supports our results: 

"Grabowska et al. also affirms the importance of re-thinking our priorities in RD surgery, giving more importance to macula-off RD repair within 3 days [36], contrary to previous articles which suggested less urgency when repairing macula-off RD [37]."

2. Regarding the fact that we considered persistent RD due to surgical failure as recurrence, we have newly acknowleged it as a limitation of our study. We have added the following explanation:

"Another limitation of the study was that due to the postoperative follow-up protocol of patients in our hospital we were not able to differentiate between patients with persistent RD related to primary anatomical failure and patients suffering RD recurrence. Therefore, we decided to include all cases with anatomical failure after primary RD surgery, based on the guidelines developed by the Australian New Zealand Society of Retinal Surgeons."

Kind regards.

Round 2

Reviewer 2 Report

The authors have addressed my concern. Thank you.